# PROFIT, a PROspective, randomised placebo controlled feasibility trial of Faecal mIcrobiota Transplantation in cirrhosis: study protocol for a single-blinded trial

Charlotte Alexandra Woodhouse,[1] Vishal C Patel,[1] Simon Goldenberg,[2] Alberto Sanchez-Fueyo,[1] Louise China,[3] Alastair O'Brien,[3] Clare Flach,[4] Abdel Douiri,[4] Debbie Shawcross[1]

For numbered affiliations see end of article.

**Correspondence to**
Dr Charlotte Alexandra Woodhouse;
charlottewoodhouse@nhs.net

## ABSTRACT

**Introduction** Patients with advanced cirrhosis have enteric bacterial dysbiosis and translocation of bacteria and their products across the gut epithelial barrier. This culminates in systemic inflammation and endotoxaemia, inducing innate immune dysfunction which predisposes to infection, and development of complications such as bleeding, sepsis and hepatic encephalopathy. This feasibility study aims to assess the safety of administering faecal microbiota transplantion to patients with cirrhosis and explore the effect of the intervention on their prognosis by achieving restoration of a healthy gut microbiome.

**Methods and analysis** A PROspective, randomised placebo controlled feasibility trial of Faecal mIcrobiota Transplantation is a single-centre, randomised, single-blinded, placebo-controlled study evaluating faecal microbiota transplantation (FMT) against placebo. Patients with advanced but stable cirrhosis with a Model for End-Stage Liver Disease score between 10 and 16 will be recruited. Twenty-four patients will be randomised to FMT plus standard of care (as per our institutional practice) and eight patients to placebo in a ratio of 3:1. Patients will be evaluated at baseline before the study intervention is administered and at 7, 30 and 90 days post-intervention to assess safety and adverse events. FMT/placebo will be administered into the jejunum within 7 days of baseline. The primary outcome measure will be safety and feasibility as assessed by recruitment rates, tolerability and safety of FMT treatment. Results will be disseminated via peer-reviewed journals and international conferences. The recruitment of the first patient occurred on 23 May 2018.

**Ethics and dissemination** Research Ethics approval was given by the London South East Research Ethics committee (ref 17/LO/2081).

**Trial registration number** NCT02862249 and EudraCT 2017-003629-13.

## INTRODUCTION

Patients with advanced cirrhosis have enteric dysbiosis with small bowel bacterial

### Strengths and limitations of this study

► This study is powered to assess feasibility and safety of administering faecal microbiota transplantation (FMT) to patients with cirrhosis, however, it is not statistically powered to assess for clinically relevant outcomes.

► This is the first study examining the effect of FMT delivered directly into the small bowel in patients with advanced cirrhosis. This trial does not involve antibiotic pretreatment in the FMT group, as has been undertaken in the USA in patients with hepatic encephalopathy.

► A PROspective, randomised placebo controlled feasibility trial of Faecal mIcrobiota Transplantation will assess instillation of FMT/placebo directly into the small bowel, as opposed to the colon, directly targeting small bowel bacterial overgrowth that is observed in cirrhosis.

► A limitation of the study is its single-blinded design, which was necessary as the FMT and placebo (saline with glycerol) are not matched.

overgrowth and translocation of bacteria and their products across the gut epithelial barrier.[1] This culminates in systemic inflammation and endotoxaemia, inducing innate immune dysfunction which predisposes to infection,[2] and development of complications such as bleeding, sepsis and hepatic encephalopathy (HE).[3] It also plays a key role in the natural history of cirrhosis by influencing the rate of progression to advanced liver disease and terminal liver failure.[4]

Using quantitative metagenomics, our group has found 75 245 genes differentially expressed between patients with cirrhosis and healthy individuals. Over 50% of these bacterial species are of buccal origin suggesting an invasion of the gut from the mouth in cirrhosis.[5] Patients

with cirrhosis also have salivary dysbiosis associated with impaired salivary defences and systemic inflammation. Salivary dysbiosis has been shown to be greater in patients with cirrhosis who developed complications necessitating hospitalisation within 90 days. Modulating the gut microbiota in patients with cirrhosis with the non-absorbable antibiotic rifaximin has been associated with improved cognitive performance and reduction in endotoxaemia in patients with cirrhosis.[6 7] Moreover, we have recently performed a multicentre retrospective study including 170 patients in which rifaximin-α therapy given for 90 days significantly (1) reduced hospital readmission rates after 3 months treatment, impacting significantly on the National Health Service resource burden and (2) reduced overall liver disease severity (as measured by the Child Pugh and Model for End-Stage Liver Disease (MELD) scores) raising the possibility that modulation of gut microbiota may significantly modify the natural history of chronic liver failure.[8]

These data constitute in our view 'proof of principle' that modifying the gut microbiota in patients with cirrhosis improves clinical outcomes. Rifaximin-α was approved by National Institute for Health and Care Excellence (NICE) for the prevention of the recurrence of overt HE in cirrhosis[9] but considerable concern remains regarding whether long-term antibiotic prescription will result in a change in bacterial function and virulence rather than a simple reduction in bacterial population and whether this may drive bacterial resistance to antibiotics in an already functionally immunocompromised population. The question was, therefore, raised as to whether directly, as opposed to indirectly modulating the gut microbiota using faeces from healthy donors may be a safer and more durable therapy. Faecal microbiota transplantation (FMT) has been licensed by NICE since 2014 for the treatment of recurrent *Clostridium difficile* infection.[10] FMT has shown promising results in clinical trials of several disease states resulting from gut dysbiosis beyond *C. difficile* infection,[10] for example, in ulcerative colitis.[11–14]

We hypothesise that in patients with advanced cirrhosis, FMT may reduce the progression of chronic liver failure including the development of jaundice, ascites, bleeding, encephalopathy, infection and organ dysfunction. Whether FMT is feasible in the setting of cirrhosis remains to be investigated. We propose conducting a feasibility trial to determine whether FMT from a healthy donor will alleviate gut dysbiosis and immune dysfunction in advanced cirrhosis.

## METHODS AND ANALYSIS
### Primary objectives
The primary objective of this study will be to assess whether stabilising gut dysbiosis with FMT in patients with advanced cirrhosis is both feasible and safe.

### Primary endpoints
The primary endpoints of the study will be twofold. To assess the feasibility of FMT as determined by the recruitment rates (including the acceptability of the intervention) and tolerability of FMT, for example, gastro-oesophageal reflux rates. The second primary outcome measure will be to assess the safety of FMT administration, including the incidence of any transmissible bacterial or viral infection that is deemed to have been acquired from the donor including *C. difficile* infection.

### Secondary objectives
The secondary objectives of the study are to provide preliminary evidence of efficacy for a larger randomised trial, with the purpose of choosing the optimal primary outcome and estimating the parameters for sample size calculation. We will also collect blood, saliva, stool and urine samples from participants to assess the stability of the transplanted gut microbiome by comparing the percentage composition of the stool microbiota on day 7, 30 and 90 with the donor microbiome. Plasma endotoxin (and endotoxin binding protein), proinflammatory and anti-inflammatory cytokine levels, bacterial DNA quantification and serum procalcitonin will be performed at 7, 30 and 90 days as will changes in faecal biomarkers (calprotectin, lactoferrin and M2-pyruvate kinase).

### Trial design
A PROspective, randomised placebo controlled feasibility trial of Faecal mIcrobiota Transplantation (PROFIT) is a single-centre study. Thirty-two patients will be recruited from outpatient clinics at King's College Hospital or from

---

**Box 1  Inclusion and exclusion criteria**

Inclusion criteria
- ▶ 18–75 years.
- ▶ Confirmed advanced cirrhosis of any aetiology with a Model for End-Stage Liver Disease[15] score between 10 and 16. The diagnosis of cirrhosis will be based on clinical, radiological or histological criteria.
- ▶ Patients with alcohol-related liver disease must have been abstinent from alcohol for a minimum of 6 weeks.
- ▶ Patients must be deemed to have a capacity to consent to the study.

Exclusion criteria
- ▶ Severe or life-threatening food allergy.
- ▶ Pregnancy or breast feeding.
- ▶ Patients treated for active variceal bleeding, infection, bacterial peritonitis, overt hepatic encephalopathy or acute-on-chronic liver failure within the past 14 days.
- ▶ Patients who have received antibiotics in the past 14 days.
- ▶ Active alcohol consumption of >20 g/day.
- ▶ Has had a previous liver transplant.
- ▶ Hepatocellular carcinoma outside of the Milan criteria.[20]
- ▶ Inflammatory bowel disease.
- ▶ Coeliac disease.
- ▶ A history of prior gastrointestinal resection such as gastric bypass.
- ▶ Patient is not expected to survive the duration of the study (90 days).
- ▶ Severe renal impairment (creatinine >150 µmol/L).
- ▶ HIV positive.
- ▶ Immunosuppression, for example, more than 2 weeks treatment with corticosteroids within 8 weeks of intervention, active treatment with tacrolimus, mycophenolate, azathioprine.

suitable inpatients on the wards. The patients will be recruited as per the inclusion and exclusion criteria listed in box 1. Patients will be randomised in a single-blinded fashion in a ratio of 3:1 FMT to placebo. Patients will be unaware of the intervention given, but investigators will not be blinded to the treatment intervention.

### Patient and public involvement

A lay reviewer/patient representative at the National Institute for Health Research reviewed the protocol. This was funded by the Research for Patient Benefit programme. Feedback was taken on board and revisions were subsequently made.

The British Liver Trust, British Society of Gastroenterology Liver Research Group and King's Liver Outpatient Advisory Group (patient group) were all involved in the study set-up and protocol development at all stages.

Results will be disseminated to study participants via the Liver Research Nurse if they indicate an interest in the study outcome.

### Patient population

The study will include all patients aged 18–75 years with a capacity to consent and a diagnosis of cirrhosis. Patients will have an MELD[15] score of between 10 and 16. The diagnosis of cirrhosis may be based on radiological, clinical or histological parameters. In the case of alcohol-related cirrhosis, patients must have been abstinent for a minimum of 6 weeks. The exclusion criteria are outlined in box 2. Anticipated recruitment is outlined in figure 1.[16]

### Consent

All participants will be asked to provide informed consent after having had the opportunity to discuss the trial with the clinician, their family and friends and having read the patient information sheet (PIS- see online supplementary file). Patients who lack capacity will not be enrolled in this study. If patients lose capacity during the trial follow-up period, an appropriate legal representative will be consulted and will provide consent to ongoing trial participation after appropriate discussion with the trials team and having read the PIS. If the legal representative does not agree to ongoing trial participation, the subject will be withdrawn from trial follow-up.

### Study intervention

FMT is prepared in a laboratory at St Thomas' Hospital in accordance with the principles of good manufacturing practice and under Manufacturing Authorisation for an Investigational Medicinal Product from the Medicines and Healthcare Products Regulatory Agency. FMT donors are healthy volunteers with no medical problems and normal body mass index. They must not be taking

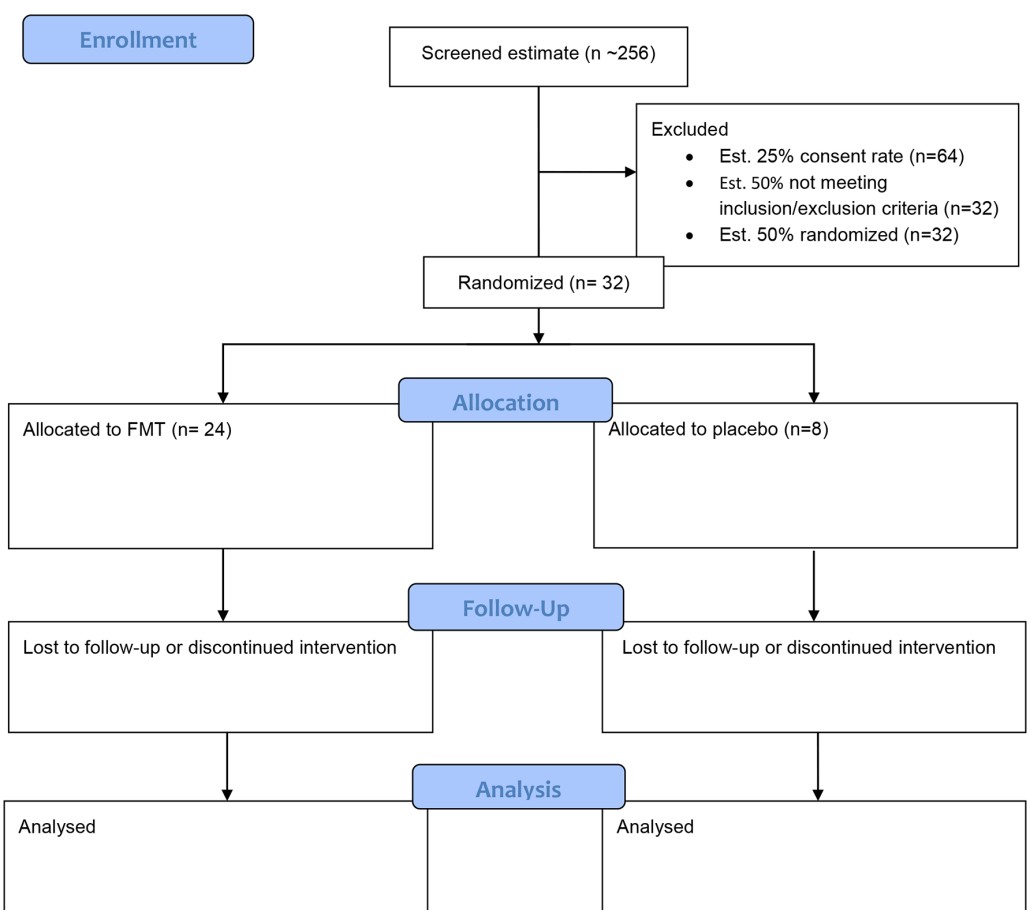

**Figure 1** Study flow chart and anticipated recruitment. FMT, faecal microbiota transplantation.

---

### Box 2 Blood and stool testing of donor faecal microbiota transplantation samples

**Blood (serology)**
► HIV 1+2 serology.
► (human T-lymphotrophic virus) HTLV I/II Ab.
► Hepatitis A IgG (and if positive IgM).
► Hepatitis B surface antigen and core antibody.
► Hepatitis C virus antibody.
► Hepatitis E.
► Syphilis.
► cytomegalovirus (CMV)/Epstein Barr Virus (EBV) IgG/M.
► *Strongyloides stercoralis* (ELISA).

**Stool**
► PCR for gastroenteritis agents (*Campylobacter, Salmonella, Shigella* and *Escherichia coli* O:157).
► Ova, cysts and parasites x3.
► *Clostridium difficile* test.
► Norovirus PCR.
► Screen for gentamicin and carbapenem-resistant Gram-negative organisms.
► Screen for methicillin resistant staphylococcus aureus (MRSA).
► Helicobacter pylori antigen.
► Entamoeba histolytica PCR.

---

any regular medications and are rigorously screened for bloodborne and enteric pathogens prior to donation. Donors undergo questionnaire screening for risk factors at baseline and again at donation to reduce the risk of transmissible infection as outlined in box 2 (full questionnaire in online supplementary materials). Stool is processed (diluted and filtered with the addition of glycerol) within 6 hours of donation and stored at −80 °C for up to 6 months (as previously described).[17] Material for FMT has full traceability from donor to recipient and aliquots of donor stool are kept for 30 years to allow further testing in the case of any adverse events. Placebo is a solution of saline with glycerol without faecal matter.

Following identification, all patients will attend a screening visit and sign the informed consent form. If deemed to be eligible, they will attend for a baseline visit with full clinical history and examination. Medication history will be recorded. Patients will complete a dietary questionnaire. Baseline samples of blood, urine, saliva and stool will be obtained. All patients will subsequently attend for a gastroscopy at which the FMT or placebo will be administered under direct visualisation into the jejunum via a nasojejunal tube. This will be performed as per the local Endoscopy Unit Protocol using topical local anaesthetic spray or midazolam sedation as per patient preference. Patients will first be asked to take bowel preparation to purge the bowel of its native bacteria. This consists of two sachets of Moviprep taken prior to gastroscopy. Patients will be monitored for side effects in the Clinical Research Facility after the procedure.

### Evaluations during and after treatment

Participants will be reassessed at 7 (±5), 30 (±7) and 90 (±7) days post-intervention. They will be reviewed in the Clinical Research Facility and undergo a physical examination, review of medications and dietary changes and adverse event monitoring. Samples of blood, urine, saliva and stool will be taken at these visits also. At the end of the 90-day follow-up period, patients will return to their usual care pathway.

### Statistical analysis

#### Sample size

This feasibility study will evaluate feasibility parameters using 95% CIs. The sample size has been proposed mainly to enable the trial to be conducted within the allocated budget and with acceptable precision for continuous outcomes. According to the simulation work by Teare *et al*,[18] even with the relatively small pilot sample size of 20, the planned studies would have at least 80% power to detect the target effect size (for continuous outcomes) more than 75% of the time. Teare *et al* recommend that 60–100 subjects are sufficient to estimate an event rate (such as recruitment rates) with acceptable precision in a feasibility study, while sample sizes between 24 and 50 have been recommended for the accurate estimation of SDs. Therefore, we have chosen a sample size of 32 patients in this trial to have reliable data on all critical parameters (including event rates) which can also be used when planning a larger intervention trial. For event rates (eg, recruitment rates) and particularly in the extreme case with lower rates, for example, 10%, we estimate a drop of precision by only 5% using our updated sample size and the minimum recommended by Teare *et al* (0.16 for 60 patients vs 0.21 for 32 patients). Figure 2 illustrates the reduction in precision of different rates when the sample size increases for binary outcomes. This sample size will also be feasible within the budget and will provide acceptable information for planning a future large clinical trial.

The sample size of 32 patients will undergo randomisation in a 3:1 ratio. This will allow the study to demonstrate

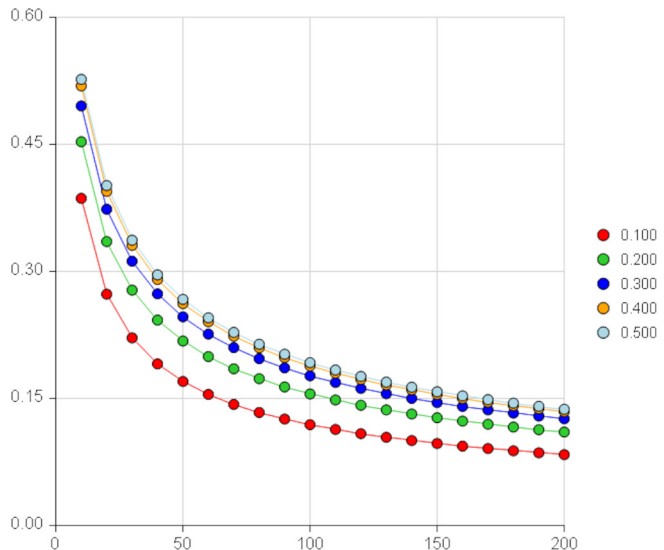

**Figure 2** CI width around one proportion (P) by sample size (N) from PASS software V.15.

**Table 1** Half-width of CIs for differences in proportions between two groups with given sample size, placebo group, p=0.5 (created with PASS software REF—PASS V.15)

| Difference | 0.1 | 0.2 | 0.3 | 0.4 | 0.5 |
|---|---|---|---|---|---|
| Width | 0.398 | 0.392 | 0.382 | 0.337 | 0.347 |

The half-width of the 95% CI of the difference in means for continuous outcome will be 0.889 SD.

the feasibility of randomising, yet providing robust evidence with respect to the feasibility of the treatment and preliminary evidence of efficacy parameters.

The following feasibility criteria have been established:

► Twenty-five per cent of screened patients will consent: 256 patients will allow the estimate of the two-sided 95% CI of the proportion of consented patients, where the distance from the observed proportion to the limit is 0.058 units.

► Fifty per cent of patients screened will fulfil inclusion/exclusion criteria for the trial: screening 64 patients will allow the estimate of the two-sided 95% CI of the proportion of patients eligible, where the distance from the observed proportion to the limit is 0.128 units.

► Eighty per cent of randomised patients will complete treatment and follow-up: 26 of 32 patients completing will allow the estimate of the two-sided 95% CI for a single proportion, where the distance from the observed proportion to the limit is 0.177 units.

Differences between the placebo and treatment group in efficacy binary outcomes will be estimated along with the two-sided 95% CI (table 1). Assuming the proportion with the outcome in the control group is 0.5, our sample size will allow estimation of the half-width of the CIs for different size of differences as indicated below. Differences of 0.4 or higher will be distinguishable from 0.

### Clinical endpoints

Clinical and safety events will be listed and summarised by the intervention group. MELD[15] scores will be calculated by visit and treatment group.

### Data synthesis, analysis and presentation

Feasibility and efficacy outcomes will be summarised using the appropriate descriptive statistics, and 95% CIs will be calculated to allow for success and go/no go decisions. Biomarker data will be preprocessed according to the established standards for each platform, and statistical analyses will be performed using non-parametric and permutation-based methods which are more appropriate for small sample sizes.

### Statistical software

Analyses will be performed using R and/or Stata V.14.2 statistical software packages.

### DISCUSSION

The PROFIT study is the first study to look at the use of FMT delivered directly into the small bowel in patients with cirrhosis. Liver transplantation is a highly selective treatment and only those most likely to benefit will be listed. Depending on blood type patients can wait up to 24 months for a liver transplant. There are even fewer options for patients who are unsuitable for transplantation.

Cirrhosis is often complicated by recurrent admissions with sepsis, variceal bleeding, ascites and HE. We know that patients with cirrhosis have enteric dysbiosis and altered gut permeability. The non-absorbable antibiotic rifaximin has proved useful in patients with cirrhosis and chronic HE, but the long-term benefits of antibiotic use are unknown and the possibility of development of bacterial resistance remains with all antibiotics. The possibility of repopulating the gut of patients with cirrhosis with healthy gut microbiota could be a potential alternative, providing a new treatment approach for these vulnerable individuals.

A group in the USA have recently published the results of the impact of FMT delivered via enema in 10 patients with cirrhosis, compared with 10 patients receiving standard of care treatment.[19] FMT was found to be safe and the patients in the treatment group did not have any admissions for HE, whereas there were six episodes of HE in the standard of care group. Four of these episodes required hospital admission. The FMT group were treated with broad-spectrum antibiotics prior to FMT administration, whereas the standard of care group were not, meaning that the effects of FMT cannot be clearly separated from the antibiotic administration. We hope that PROFIT will add to the knowledge base of the use of FMT in cirrhosis and will provide an accurate assessment of the safety of FMT. We plan to treat the FMT and control groups identically, aside from the administration of FMT to accurately assess its impact, without the confounding factor of antibiotic treatment.

A limitation of our study is the single-blinded design. This was selected due to the inherent difficulties in preparing a matched placebo, without introducing substances that may upset the delicate gut microbiota. The IMP will be delivered by the trial investigators, so it has not been possible to blind the clinicians in this study. Patients and the trial statistician will be blinded.

The study is not powered to detect differences in clinical outcomes, but may provide evidence for markers relating to clinical outcomes that could be studied in a larger randomised controlled trial.

### Trial monitoring groups
#### Trial steering committee

This group will oversee the running of the trial and discuss any issues that may arise throughout the process of recruitment and follow-up of patients. The group will be chaired by an independent clinician. Investigators will report to the group on a regular basis. The

data monitoring committee (DMC) will inform the trial steering committee (TSC) if there are any issues raised from its discussions.

## Data monitoring committee

This is an entirely independent group that analyses interim data, to determine whether or not the trial is safe to continue. It monitors adverse events and adverse reactions and reacts to any issues and directs the TSC as to whether or not the trial should continue. The DMC undertakes interim statistical analysis using an independent statistician to ensure the ongoing safety and integrity of the trial. The members of this committee are independent of the trial, but will be experienced clinicians with expertise in clinical trials.

## ETHICS AND DISSEMINATION

The results of the trial will be analysed and published in a peer-reviewed journal and disseminated at international conferences.

**Author affiliations**
[1]James Black Centre, School of Immunology & Microbial Sciences, Faculty of Life Sciences & Medicine, King's College London, London, UK
[2]Department of Microbiology, Guy's & St Thomas' NHS Foundation Trust, London, UK
[3]Division of Medicine, University College London, London, UK
[4]School of Population Health & Environmental Sciences, Faculty of Life Sciences & Medicine, King's College London, London, UK

**Acknowledgements**  We would like to thank the clinicians involved, King's College Hospital R&D Department, the NIHR, patient advisers and lay reviewers for their contributions in the set-up of the PROFIT trial. We would also like to acknowledge Liz Allen, our QP, for her contribution to the MHRA approval process.

**Contributors**  CAW and DS wrote the manuscript, with assistance from SG, VCP and AS-F. AD and CF prepared the statistical analysis plan. LC and AO advised on study analyses and the format of the protocol for submission. SG provided the expert advice on FMT and has set up the FMT service at GSTT, which will supply the FMT to the study. All authors reviewed and edited the manuscript prior to submission.

**Funding**  This work was supported by the NIHR, grant number PB-PG-0215-36070. This paper presents independent research funded by the National Institute for Health Research (NIHR) under its Research for Patient Benefit (RfPB) Programme (grant reference number PB-PG-0215-36070). The PROFIT trial is co-sponsored by Kings' College Hospital and Kings' College London.

**Competing interests**  DS has received fees from Norgine, Falk and Shinogi, outside the submitted work. SG reports grants and personal fees from Astellas, personal fees from MSD, personal fees from Pfizer, personal fees from Shinogi, outside the submitted work.

**Patient consent for publication**  Not required.

**Ethics approval**  The ethical permission has been given by the South-East Research Ethics Committee, REC reference 17/LO/2081, IRAS project ID 197237

**Provenance and peer review**  Not commissioned; externally peer reviewed.

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
