## [Reviewer comments · BMJ Open]

ARTICLE DETAILS

TITLE (PROVISIONAL)	PROFIT: A PROspective, randomised placebo controlled feasibility trial of Faecal mIcrobiota Transplantation in cirrhosis: study protocol for a single-blinded trial
AUTHORS	Woodhouse, Charlotte; Patel, Vishal; Goldenberg, Simon; Sanchez Fueyo, Alberto; China, Louise; O'Brien, Alastair; Flach, Clare; Douiri, Abdel; Shawcross, Debbie

VERSION 1 – REVIEW

REVIEWER	Benjamin Mullish Division of Integrative Systems Medicine and Digestive Disease, St Mary's Hospital Campus, Imperial College London
REVIEW RETURNED	12-May-2018

GENERAL COMMENTS	Main document: This is a very nicely-written, clear protocol, but I have a few comments: Page 33 line 33 - Is it reasonable to call this a 'single blinded' study? Given the placebo that is being used (saline + glycerol) will likely be so different in appearance to FMT – and the theoretical possibility for the patient to see the syringes of treatment during its endoscopic administration – it might be argued that there is no blinding here. Will syringes be covered/ disguised? Would autonomous FMT (i.e. retransplantation with FMT prepared from the recipient's stool) not be a reasonable comparator arm? Page 4 line 26 - FMT is universally still viewed as an experimental therapy – albeit one with cautious optimism – in the treatment of ulcerative colitis, and has not been used in a randomised trial to treat Crohn's disease. References to FMT as treatment of IBD should be more circumspect. Page 5 line 35 – I do not think there is a comment about the use of lactulose; is this permitted? One proposed mechanism of its use clearly relates to pH change and colonic bacterial profile. Page 6 - Should recent use of probiotics be one of the exclusion criteria? Should patients at risk of aspiration pneumonitis (previous stroke, severe GORD, previous ENT surgery, etc) also be excluded? Page 7 – It seems an oversight not to mention health questionnaire screening in more detail, rather than just listing infections. Would you accept a donor who had had antibiotics last week? Or a donor with cirrhosis?
---

	Page 8 line 20 - Are there not concerns about an increased risk of aspiration in patients who are administered sedation for the gastroscopy? There has been at least one fatal case of aspiration pneumonitis described in the literature after endoscopic small bowel FMT administration. General - Why have the timepoints of 7, 30 and 90 days for assessment post-intervention been selected? It might be expected that patients may die or undergo liver transplant prior to completion of the protocol – what will happen in this case? Will additional participants be recruited if so? Consent Form: A study-specific PIS for donor is referred to in the protocol but no PIS/ consent form provided here. Presumably donors are also being asked to donate biofluids for comparison with the FMT recipients? Investigators may wish to consider addition of reference to intellectual property in a consent form. For example, investigators may identify that particular gut microbial community members from a certain donor associate with favourable clinical and/ or immunological outcomes, and may wish to culture these organisms and use them as an alternative to FMT in future studies. If so, does 'ownership' of these microorganisms belong to the donor or to the investigators? Participant Information Sheet: There is variable spacing, font/ font size and text justification throughout the PIS (see 'what if something goes wrong?' section for example). Some phrases could be perceived as jargon, e.g. 'well tolerated', 'aseptic conditions', etc. There should be a clearer lay explanation of what 'faecal microbiota transplantation' is, since this may clearly influence patient choice about participation. There is no specific explanation about taking a stool sample from a screened donor and formulating a liquidised bacterial suspension from it. Change 'clostridium difficile' to Clostridium difficile throughout. As described above, it is misleading to say 'FMT...has been used more widely in other conditions such as patients with....inflammatory bowel disease....it is extremely successful in these patients. With regards to risks of taking part – the beginning of the information sheet mentions 'leaky gut' and gut microbial translocation into the portal circulation. Is this not a theoretical complication of FMT that is worth describing? Protocol: Overall, this is a very clear and comprehensive protocol, although I have some outstanding questions. As above – is this really a single blinded study? As above – do the exclusion criteria need revision?
--	--

	Page 18 – once again, this should perhaps be more circumspect regarding the success of FMT in the treatment of IBD. Page 22 – References are made to collecting urine and faeces for metabonomic/ bile acid analysis, but no metabonomic analysis is included within the ‘Mechanistic Outcomes’. Investigators should state this aim here. Is there merit in exploring beyond purely bile acid profiling, e.g. faecal water NMR to assess for nitrogenous metabolites? GC MS for short chain fatty acids? Page 22 – the investigators make several references to gut leak throughout. Is there no merit to investigating whether the FMT will affect gut leak, e.g. via change in bacterial protease profile? Page 32 – which samples will donors be asked to give? Will that be purely the stool sample for FMT? Page 42 – should PPI and metoclopramide be considered prior to administration? Although this protocol involves small bowel administration, it is feasible for gastric content to be carried on the endoscope tip from the stomach into the small bowel during gastroscopy. Page 42 – as above, are there not concerns about an increased risk of aspiration in patients who are administered sedation for the gastroscopy?
--	--

REVIEWER	Patrizia Kump Division of Gastroenterology, Department of Internal Medicine, Medical University Graz, Austria
REVIEW RETURNED	05-Jul-2018

GENERAL COMMENTS	I read the protocol of the PROFIT (prospective, randomised placebo controlled feasibility trial of fecal microbiota transplantation in cirrhosis) written by Woodhouse et al with interest. It has potential to give important insights for future trials in this field. I do, however, have a few thoughts I would like to share. 1) Polyethylene Glycol will be used pretransplant to prepare recipients, as it has been recommended in the European Guidelines (Cammarota et al Gut 2017). However, in hepatic encephalopathy (HE) it has beneficial effects similar or even superior to lactulose (Rahimi R et al, JAMA Intern Med 2014). As HE will not be an exclusion criteria, it should be measured by scores and ammonia levels prior and after bowel lavage as well as after FMT. 2) The rates of severe adverse events (SAEs) seem to be higher after FMT via the upper GI tract. Cases of aspiration pneumonia due to nausea and vomiting have been reported. The authors plan to apply 200mls of prepared donor stool or placebo into the duodenum via gastroscopy without a nasoduodenal tube. The amount of fecal transplant seems to be relatively high to be administered without nasoduodenal tube. A tube provides the possibility of an infusion over a longer time interval and might reduce the risk of SAEs. 3) Bajaj JS reported already feasibility and safety of FMT via lower GI tract in liver cirrhosis and HE. As the route of application has been shown to be safe, FMT via the lower GI tract but not via the upper GI tract should also be preferred in the PROFIT study. In
---

	any case, that would improve the comparability of these two pilot studies.
REVIEWER	Perttu Arkkila Helsinki University Hospital, Finland
REVIEW RETURNED	21-Jul-2018
GENERAL COMMENTS	Well designed and important study. Few remarks: -Is only single donor used? Are there any special additional inclusion criteria for the donor. The list of inclusion criteria is the same as for the FMT procedure for rCDI. For example should donors be normal weight, no history of NAFLD or type 2 diabetes, liver disease in the family etc. -What to do if side effect appear, for example GERD of post infectious IBS. Is treatment with PPI or antibiotics allowed during follow up period. -Are esophageal varices Gr II-III contraindication for FMT? -Regarding the risk for aspiration, patients should be able to be at up-right position soon after gastroscopy. -Concomitant medication: probiotics? PPI? Exclusion criteria or stable dose should be mentioned.

VERSION 1 – AUTHOR RESPONSE

Reviewer: 1 Reviewer Name: Benjamin Mullish Institution and Country: Division of Integrative Systems Medicine and Digestive Disease, St Mary's Hospital Campus, Imperial College London Please state any competing interests or state 'None declared': None declared Please leave your comments for the authors below Main document: This is a very nicely-written, clear protocol, but I have a few comments:

Q: Page 33 line 33 - Is it reasonable to call this a 'single blinded' study? Given the placebo that is being used (saline + glycerol) will likely be so different in appearance to FMT – and the theoretical possibility for the patient to see the syringes of treatment during its endoscopic administration – it might be argued that there is no blinding here. Will syringes be covered/ disguised? Would autonomous FMT (i.e. retransplantation with FMT prepared from the recipient's stool) not be a reasonable comparator arm?

A: Thank you for your comments Ben. Blinding was discussed with our ethics committee and in fact they wanted us to make PROFIT double blind, but unfortunately due to personnel constraints this was not possible as we have been giving the treatment ourselves. We have put considerable thought into preventing the subjects from seeing the IMP. We have ensured it is kept in a sealed, opaque container until it is ready to be used. Due to positioning of the trolley and as the IMP is administered via an NJ tube, remote from the patient's face, the syringes are handled behind the patient and we have been very careful to make sure they are not seen. The patients are all sedated, which in practice has meant that they have not remembered the procedure. PROFIT is very much a safety and feasibility trial. Autologous stool transplantation for the placebo arm could be considered in a subsequent trial, however we do not have ethical approval for this and we have already begun recruitment and treatment with the saline/glycerol placebo solution.

Q: [L L L L] [SEP] [SEP] Page 4 line 26 - FMT is universally still viewed as an experimental therapy – albeit one with cautious optimism – in the treatment of ulcerative colitis, and has not been used in a randomised trial to treat Crohn’s disease. References to FMT as treatment of IBD should be more circumspect.

A: I will amend the wording of this to clarify, thank you. [L L L L] [SEP] [SEP]

Q: Page 5 line 35 – I do not think there is a comment about the use of lactulose; is this permitted? One proposed mechanism of its use clearly relates to pH change and colonic bacterial profile.

A: Whilst lactulose does have effects on the colonic pH, potentially altering gut microbiota, as you have mentioned, we have adopted a practical approach and not excluded those taking lactulose from participation in the trial as it is standard of care in most patients with CLD. PPI use, which is also common in our cohort, also disrupts the gut microbiota (1, 2) as does metformin treatment (3). Maier et al. have recently published an interesting article in which they tested over 1000 commonly used drugs against 40 known commensal bacteria and found that 24% of the drugs with human targets, including members of all therapeutic classes, inhibited the growth of at least one strain in vitro (4). We have therefore adopted a pragmatic approach, given the myriad of pharmaceuticals that can impact upon gut microbiota, and only excluded those on antibiotics in the last fourteen days (as this may impact directly on the viability of FMT) and those on immunosuppressive agents due to the theoretical risks of infection in an already functionally immunocompromised cohort.

[L L] [SEP] Q: Page 6 - Should recent use of probiotics be one of the exclusion criteria? Should patients at risk of aspiration pneumonitis (previous stroke, severe GORD, previous ENT surgery, etc) also be excluded?

A: We have not specifically mentioned probiotic use in the exclusion criteria as probiotic use in our patient cohort is uncommon. As you will be aware probiotics are not subject to the same regulation as medications. We have included a dietary questionnaire to try to address the impact of food-stuffs that may have a prebiotic effect. As regards aspiration risk, we do not have many patients in our cohort with these comorbidities, therefore this has not been addressed as a specific exclusion criteria. This is a reasonable suggestion, however I do not think that previous stroke would necessarily exclude a patient from participation, if they were well enough to be living independently at home and attending the Liver Outpatient clinic and had capacity to consent. If they were eating normally and not PEG fed then we would take into account any SALT assessment when assessing their aspiration risk and potential trial involvement.

Q: [L L] [SEP] Page 7 – It seems an oversight not to mention health questionnaire screening in more detail, rather than just listing infections. Would you accept a donor who had had antibiotics last week? Or a donor with cirrhosis?

A: This was edited for brevity- I will expand to state that all donors are healthy volunteers with normal BMI and are not on any medications. A full personal, travel and sexual history is obtained from the donors to exclude potential risk factors for transmissible infection. As a safety and feasibility trial the main focus was on infection so this was the main focus of the detail in this section, but I will expand to address your comments.

Q: [L L L L] [SEP] [SEP] Page 8 line 20 - Are there not concerns about an increased risk of aspiration in patients who are administered sedation for the gastroscopy? There has been at least one fatal case of aspiration pneumonitis described in the literature after endoscopic small bowel FMT administration.

A: Patients are fully informed about the risks of aspiration during endoscopy and as you will be aware, aspiration is a potential risk in all upper GI endoscopy (even when patients have been nil by mouth for

the requisite six hours as gastric acid and small bowel secretions and even saliva/flushed saline may be aspirated.) Patients having lower GI endoscopy with sedation are also at risk of aspiration due to the respiratory depression conferred by sedation, so this is not unique to the upper GI route of delivery. Baxter et al.(5) reported a death from respiratory failure of a patient who received FMT via the colonic route for CDI (Kelly CR, Ihunnah C, Fischer M, et al. Fecal microbiota transplant for treatment of Clostridium difficile infection in immunocompromised patients. Am J Gastroenterol 2014;109:1065-1071). Given that aspiration can occur in both upper and lower GI endoscopy, we have opted for the Upper GI route as this is where the dysbiosis is thought to be at its peak, due to altered small bowel motility and small bowel bacterial overgrowth that occurs in cirrhosis. Instillation in the small bowel also facilitates peristaltic transport of the FMT to the colon. Our FMT is prepared as a 200mL solution. A smaller volume was considered, but this made the solution too viscous to pass easily down the NJ tube. 200mL has been set as the minimum possible volume for adequate dissolution of the 50g stool volume, which also allows easy passage of the solution via the NJ tube. This volume was selected based upon experience administering FMT via NJ to patients with CDI at St Thomas' Hospital. Patients are sat upright immediately post IMP administration and monitored closely in recovery. Sedation is vital as passing the volume of FMT takes time, due to its viscosity, as you may have experienced in your own practice. If not appropriately sedated patients are liable to retch/vomit and potentially put themselves at increased risk of aspiration. In the few patients we have treated so far this has been reasonably well tolerated and tolerability has improved with the use of CO2 insufflation, instead of air.

Q: [SEP]General - Why have the time-points of 7, 30 and 90 days for assessment post-intervention been selected? It might be expected that patients may die or undergo liver transplant prior to completion of the protocol – what will happen in this case? Will additional participants be recruited if so?

A: The inclusion/exclusion criteria state that those patients not expected to survive the 90 duration of the trial will not be enrolled. In practice patients with a MELD of 10-16 are sufficiently unwell as to require intervention/consider enrolment in a clinical trial, but not so unwell as to be immediately at risk of death. We have recruited patients who have recently been assessed for transplantation, but the transplant assessment period is not so rapid that patients are at risk of being transplanted prior to the end of the follow up period. The introduction of the new 'transplant benefit score' has superseded the previous organ allocation system (which was based on proximity of donors to recipients) so waiting times are less predictable, but previously patients with blood group O could wait 18-24 months on the transplant list. 90 days has been selected as a practical short term endpoint to assess feasibility of FMT as an intervention in this cohort.

Q: [SEP][SEP]Consent Form: [SEP]A study-specific PIS for donor is referred to in the protocol but no PIS/consent form provided here. Presumably donors are also being asked to donate bio-fluids for comparison with the FMT recipients?

A: The donor information sheet used for clinical practice at GSTT for the CDI service has been used for PROFIT. This allows use of stool for both PROFIT and CDI treatment. The only bio-fluid obtained from the donor for the purposes of the PROFIT study is stool. An aliquot of which is stored for comparison of donor microbiome with recipient for trial purposes. Stool is also stored for retrospective testing in the unlikely event of a transmissible infection. Donor stool and blood are tested, as previously described, for exclusion of transmissible pathogens, but this is not trial specific. A serum saved sample is stored for the recipient also. [SEP][SEP]

Q: Investigators may wish to consider addition of reference to intellectual property in a consent form. For example, investigators may identify that particular gut microbial community members from a certain donor associate with favourable clinical and/ or immunological outcomes, and may wish to

culture these organisms and use them as an alternative to FMT in future studies. If so, does 'ownership' of these microorganisms belong to the donor or to the investigators?

A: The donor microbial patterns will be compared to patients with cirrhosis, so are unlikely to have bearing on healthy donors. Our patient facing documents have all been reviewed by the REC and R&D department locally and have been approved for trial use.

Q: Participant Information Sheet: There is variable spacing, font/ font size and text justification throughout the PIS (see 'what if something goes wrong?' section for example).

A: This was a formatting error, apologies. This PIS has been approved by the REC.

Q: Some phrases could be perceived as jargon, e.g. 'well tolerated', 'aseptic conditions', etc.

A: Again, our documents have been reviewed by the REC and no they did not recommend any changes. We cannot therefore alter the approved documents without ethical approval.

Q: There should be a clearer lay explanation of what 'faecal microbiota transplantation' is, since this may clearly influence patient choice about participation. There is no specific explanation about taking a stool sample from a screened donor and formulating a liquidised bacterial suspension from it.

A: This may introduce more jargon- in practice the patients recruited so far (who are from differing educational and social backgrounds) have all understood that the FMT is obtained from healthy donors and administered at gastroscopy, therefore they are not required to drink it/take it in tablet form and they are all aware that it is a single treatment. The format/consistency of this treatment has not been queried. These documents have been REC approved.

Q: Change 'clostridium difficile' to Clostridium difficile throughout.

A: Does this comment refer to the patient information sheet or the article itself? I have double checked the article and it appears to be correct?

Q: As described above, it is misleading to say 'FMT...has been used more widely in other conditions such as patients with...inflammatory bowel disease....it is extremely successful in these patients. With regards to risks of taking part – the beginning of the information sheet mentions 'leaky gut' and gut microbial translocation into the portal circulation. Is this not a theoretical complication of FMT that is worth describing?

A: We have described the risk of infection, which this refers to.

Q: Protocol: Overall, this is a very clear and comprehensive protocol, although I have some outstanding questions. As above – is this really a single blinded study?

A: –yes, please see comments above. As above – do the exclusion criteria need revision? –see above. Page 18 – once again, this should perhaps be more circumspect regarding the success of FMT in the treatment of IBD. -Noted

Q: Page 22 – References are made to collecting urine and faeces for metabonomic/ bile acid analysis, but no metabonomic analysis is included within the 'Mechanistic Outcomes'. Investigators should state this aim here. Is there merit in exploring beyond purely bile acid profiling, e.g. faecal water NMR to assess for nitrogenous metabolites? GC MS for short chain fatty acids?

A:- the specific assays/metabolites will be confirmed once we have the samples available for processing. The NIHR funding was provided for the safety/feasibility trial and not for the mechanistic outcomes, however we have been allowed to collect samples for these outcomes. These will be addressed once all samples have been collected and we are ready to process them. [REDACTED]

Q: Page 22 – the investigators make several references to gut leak throughout. Is there no merit to investigating whether the FMT will affect gut leak, e.g. via change in bacterial protease profile?

A: PROFIT is a safety and feasibility study, so has been designed as such. We plan to perform metagenomic analysis on stool and saliva from patients (and donor stool) to assess donor/recipient microbial profiles. Depending on what signals are detected (and if we have sufficient resources) it could be interesting to address mechanisms of gut permeability. [REDACTED]

Q: Page 32 – which samples will donors be asked to give? Will that be purely the stool sample for FMT?

A: Just stool for the purposes of FMT. Blood/stool will be tested for transmissible infection to allow the donor to donate to the general FMT service at GSTT. [REDACTED]

Q: Page 42 – should PPI and metoclopramide be considered prior to administration? Although this protocol involves small bowel administration, it is feasible for gastric content to be carried on the endoscope tip from the stomach into the small bowel during gastroscopy.

A:– we have deliberately avoided PPI and metoclopramide administration. PPI has been avoided due to its impact upon gut microbial populations. Metoclopramide has not been used due to the potential for enhance transit of the FMT, which may be lost prior to engraftment if it passes too rapidly through the UGI tract.

Q: [REDACTED] Page 42 – as above, are there not concerns about an increased risk of aspiration in patients who are administered sedation for the gastroscopy? Please see comments above. The risk of aspiration is not unique to gastroscopy and has been reported in colonoscopic administration with sedation. [REDACTED]

Reviewer: 2 [REDACTED] Reviewer Name: Patrizia Kump [REDACTED] Institution and Country: Division of Gastroenterology, Department of Internal Medicine, Medical University Graz, Austria [REDACTED] Please state any competing interests or state 'None declared': None declared [REDACTED] Please leave your comments for the authors below [REDACTED] I read the protocol of the PROFIT (prospective, randomised placebo controlled feasibility trial of fecal microbiota transplantation in cirrhosis) written by Woodhouse et al with interest. It has potential to give important insights for future trials in this field. I do, however, have a few thoughts I would like to share. [REDACTED]

Q:1) Polyethylene Glycol will be used pretransplant to prepare recipients, as it has been recommended in the European Guidelines (Cammarota et al Gut 2017). However, in hepatic encephalopathy (HE) it has beneficial effects similar or even superior to lactulose (Rahimi R et al, JAMA Intern Med 2014). As HE will not be an exclusion criteria, it should be measured by scores and ammonia levels prior and after bowel lavage as well as after FMT.

A:– Thank you for your comments, Dr Kump, this has been recognised and we are recording the Westhaven HE scores for all patients at screening, baseline, day 7, 30 and 90 as well as monitoring plasma ammonia levels to try to assess for any changes pre- and post IMP administration. Both the control and FMT arms receive Moviprep, which we hope may help to address its potential influence on the outcome.

Q: 2) The rates of severe adverse events (SAEs) seem to be higher after FMT via the upper GI tract. Cases of aspiration pneumonia due to nausea and vomiting have been reported. The authors plan to apply 200mls of prepared donor stool or placebo into the duodenum via gastroscopy without a nasoduodenal tube. The amount of fecal transplant seems to be relatively high to be administered without nasoduodenal tube. A tube provides the possibility of an infusion over a longer time interval and might reduce the risk of SAEs.

A: the IMP is instilled via an NJ inserted at OGD, under direct vision. We have not left the tube in situ as this is not required and may be displaced, increasing the risk of SAEs. Aspiration risk is not unique to the upper GI route of administration and may occur in patients sedated for colonoscopy. 200mL has been used in the treatment of CDI at St Thomas' hospital via the upper GI route also and is the minimum volume feasible for easy passage of FMT down the fine bore NJ tube.

Q: 3) Bajaj JS reported already feasibility and safety of FMT via lower GI tract in liver cirrhosis and HE. As the route of application has been shown to be safe, FMT via the lower GI tract but not via the upper GI tract should also be preferred in the PROFIT study. In any case, that would improve the comparability of these two pilot studies.

A: Bajaj et al. have shown FMT (with antibiotic pre-treatment) to be safe in cirrhotic subjects. Our trials are not directly comparable as the FMT group in their trial also received five days of broad spectrum antibiotics, whereas the control group did not receive antibiotics. They used a single donor with a 'favourable' donor profile. Mechanistically, lower GI administration of FMT does not make sense- delivery via enema requires ability to retain the fluid and does not target the small bowel bacterial overgrowth seen in cirrhosis. Instillation of FMT into the sigmoid colon would have difficulty addressing small bowel bacterial overgrowth. Upper GI administration allows direct targeting of the small bowel and also allows longer retention as the material passes through the GI tract, with the aim of enhancing engraftment of the donor microbiome. FMT delivered via the upper GI tract will reach the colon and therefore treats the whole of the GI tract. This route of administration has been approved by the REC after careful consideration and support from Jasmohan Bajaj as an independent expert. We are cognisant to the risks of aspiration, but this can occur in colonoscopy when sedation is given. All care is taken to reduce this risk and patients are consented for the risk of aspiration. Patients are sat upright immediately post procedure and observed closely for a minimum of two hours post endoscopy.

Reviewer: 3 Reviewer Name: Perttu Arkkila Institution and Country: Helsinki University Hospital, Finland Please state any competing interests or state 'None declared': None declared Please leave your comments for the authors below Well designed and important study. Few remarks:

Q: Is only single donor used? Are there any special additional inclusion criteria for the donor. The list of inclusion criteria is the same as for the FMT procedure for rCDI. For example should donors be normal weight, no history of NAFLD or type 2 diabetes, liver disease in the family etc.

A: –thank you for your comments, Dr Arkkila. Unlike the study from Bajaj et al in the USA, we have several donors, all of whom are of normal BMI and are not on any medications. None of the donors have any medical problems. This was abbreviated for length, but I will expand for clarity. I believe that the donor used in the Bajaj study had an elevated BMI and subsequently developed NAFLD. I have added the donor screening questionnaire to the supplementary materials for additional information.

Q-What to do if side effect appear, for example GERD or post infectious IBS. Is treatment with PPI or antibiotics allowed during follow up period.

A: PPI and Antibiotic treatment is allowed in the follow up period if clinically indicated. Where possible these would be avoided, but we would not withhold antibiotics if there is a clinical need. Many of these patients are already on PPIs, but we are well aware of the impact of PPI upon gut microbiota, so this would be introduced only if absolutely necessary.

Q: -Are esophageal varices Gr II-III contraindication for FMT?

A: this will be assessed on a case by case basis. We will give FMT if safe to do so, therefore well covered grade II varices should not preclude treatment, but if grade III varices are seen or stigmata of recent haemorrhage are noted it would not be safe to proceed if these would have to be treated. Therefore the varices would need to be treated prior to IMP treatment and this would have to be postponed. -Regarding the risk for aspiration, patients should be able to be at up-right position soon after gastroscopy.

Q: -Concomitant medication: probiotics? PPI? Exclusion criteria or stable dose should be mentioned. Please see above comments. We are recording medication use at each visit.

1. Bajaj JS, Acharya C, Fagan A, White MB, Gavis E, Heuman DM, et al. Proton Pump Inhibitor Initiation and Withdrawal affects Gut Microbiota and Readmission Risk in Cirrhosis. *Am J Gastroenterol.* 2018.
2. Jackson MA, Goodrich JK, Maxan ME, Freedberg DE, Abrams JA, Poole AC, et al. Proton pump inhibitors alter the composition of the gut microbiota. *Gut.* 2016;65(5):749-56.
3. Wu H, Esteve E, Tremaroli V, Khan MT, Caesar R, Manneras-Holm L, et al. Metformin alters the gut microbiome of individuals with treatment-naive type 2 diabetes, contributing to the therapeutic effects of the drug. *Nat Med.* 2017;23(7):850-8.
4. Maier L, Pruteanu M, Kuhn M, Zeller G, Telzerow A, Anderson EE, et al. Extensive impact of non-antibiotic drugs on human gut bacteria. *Nature.* 2018;555(7698):623-8.
5. Baxter M, Colville A. Adverse events in faecal microbiota transplant: a review of the literature. *J Hosp Infect.* 2016;92(2):117-27.

VERSION 2 – REVIEW

REVIEWER	Dr Benjamin Mullish Liver Unit/ Division of Integrative Systems Medicine and Digestive Disease, Imperial College London
REVIEW RETURNED	19-Sep-2018

GENERAL COMMENTS	I have reviewed the responses to my previous comments and those of other reviewers, and they are entirely appropriate. In my opinion, this submission is clear, thorough and well-written. Just a few tiny points for consideration (perhaps could be considered when proofs arrive if accepted):  -The use of the term 'dysbiosis' in the manuscript is debatable for its appropriateness (see: https://www.nature.com/articles/nmicrobiol2016228); authors may wish to consider an alternative way to describe perturbation of the structure and/or function of the gut microbiota. -The other reviewers share my concern about potential safety issues relating to administration of 200ml of FMT via the upper GI route and the use of sedation; the response has been that there is a risk of aspiration associated with both upper and lower GI route
---

	administration. However, the particular concern about upper GI administration and sedation relates to aspiration of the FMT itself (any aspiration after sedation following lower GI administration is likely a very modest amount of residual gastric secretions). This should be borne in mind.
REVIEWER	Patrizia Kump Dep. of Gastroenterology and Hepatology, Medical University Graz
REVIEW RETURNED	06-Oct-2018
GENERAL COMMENTS	The PROFIT study is a well designed trial that will give important insight for future research in this field. All my concerns have been addressed adequately by the authors.

VERSION 2 – AUTHOR RESPONSE

Many thanks for the reviewers' comments in response to our replies.

In response to Dr Mullish's concerns regarding the safety of 200mL of FMT being delivered into the upper GI tract we are acutely aware of the the risk of aspiration of FMT and have taken the utmost care to ensure that the material is delivered as safely as possible. Cirrhotic subjects have been shown to have small bowel bacterial overgrowth. As a result we have chosen to deliver the FMT directly into the small bowel to deliver the therapy directly to its site of action. We anticipate that this will allow longer retention of the FMT as compared to lower GI delivery and allows treatment of the small and large bowel simultaneously as the FMT moves through the GI tract to engraft in the colon.

We have so far treated seven patients and found that the volume delivered is well tolerated. 200mL is the minimum volume possible to ensure that we are physically able to pass the liquid down the NJ tube- the solution is very viscous and if the volume were to be reduced we would not practically be able to pass it via an NJ. The endoscope is introduced into the distal duodenum/proximal jejunum and the NJ tube passed even further down to ensure delivery is as direct to the small bowel as possible. In practice the FMT is rapidly carried down via peristalsis and there is very little reflux. Post-pyloric delivery again reduces the risk of aspiration. Whilst we appreciate the inherent risk of sedation in a patient with an unprotected airway, we have given all of the treated patients conscious sedation to ensure their comfort and to reduce retching, which might cause reflux of FMT. Switching to CO2 insufflation has also improved patient comfort and reduced belching. FMT is delivered with the patient at 45 degrees head-up tilt and patients are sat up immediately after endoscopy and monitored closely in endoscopy recovery for 2h post- procedure. We are cognisant of the risks of aspiration of FMT, but firmly believe this is the preferred route of administration to achieve the desired therapeutic effect.

With regards to Dr Mullish's concerns regarding the use of the term 'dysbiosis' we have chosen to use this term as it was used in the protocol and is frequently deployed in research relating to the microbiome. It is a convenient descriptor to relate changes in bacterial composition associated with disease, however the use of this term also sums up the complexities of research in the microbiome. With the advent of next generation sequencing, our knowledge of the composition of the gut microbiome has exploded, however our terminology has lagged behind these technological leaps. We do not truly understand what a 'healthy' microbiome is and therefore to describe as 'dysbiotic' any

microbiome associated with disease is a vast over-simplification. We are moving away from a merely descriptive analysis of the gut microbiota to a more complex understanding of the interactions between gut microbes. Perhaps our terminology needs to change as our knowledge develops, but for brevity and ease of description we have used the term dysbiosis, but as he suggests this is merely the tip of the iceberg.